# Promoting Laparoscopic Anterior Approach for a Very Low Presacral Primary Neuroendocrine Tumor Arising in a Tailgut Cyst

**DOI:** 10.3390/healthcare10050805

**Published:** 2022-04-26

**Authors:** Maria Michela Di Nuzzo, Carlo De Werra, Mirella Pace, Raduan Ahmed Franca, Maria D’Armiento, Umberto Bracale, Ruggero Lionetti, Michele D’Ambra, Armando Calogero

**Affiliations:** 1Department of Public Health, University of Naples Federico II, 80138 Naples, Italy; dewerra@unina.it (C.D.W.); maria.darmiento@unina.it (M.D.); umberto.bracale@gmail.com (U.B.); ruggero.lionetti@unina.it (R.L.); michele.dambra@unina.it (M.D.); armando.calogero2@unina.it (A.C.); 2Department of Biomorfological and Functional Sciences, University of Naples Federico II, 80138 Naples, Italy; pacemirella83@virgilio.it (M.P.); raduanahmed.franca@unina.it (R.A.F.)

**Keywords:** tailgut cyst, laparoscopy, neuroendocrine tumor

## Abstract

Background: Tailgut cysts are rare congenital lesions that develop in the presacral space. As they can potentially conceal primary neuroendocrine tumors, surgical excision is suggested as the treatment of choice. However, specific management guidelines have yet to be developed. A posterior approach is usually preferred for cysts extending to the third sacral vertebral body. Conversely, a transabdominal approach is preferred for lesions extending upward to achieve an optimal view of the surgical field and avoid injuries. Case report: Here, we report a case of a 48-year-old man suffering from perianal pain and constipation. Digital rectal examination and magnetic resonance imaging revealed a presacral mass below the third sacral vertebral body. A laparoscopic transabdominal presacral tumor excision was performed. The final histological diagnosis was a rare primary neuroendocrine tumor arising from a tailgut cyst. The postoperative course was uneventful, and no signs of recurrence were observed at the six-month follow-up. Conclusions: This study may help establish more well-grounded recommendations for the surgical management of rectal tumors, demonstrating that the laparoscopic transabdominal technique is safe and feasible, even for lesions below the third sacral vertebral body. This approach provided an adequate view of the presacral space, facilitating the preservation of cyst integrity, which is essential in cases of malignant pathologies.

## 1. Introduction

A tailgut cyst or retrorectal cystic hamartoma is a rare congenital lesion [1] that develops in the presacral space from the persistent remnants of the embryonic hindgut as a result of incomplete involution during embryogenesis [2]. Malignant transformation can occur in tailgut cysts, resulting in neuroendocrine tumors (NETs), carcinomas, or adenocarcinomas [1]. Therefore, complete resection is recommended. Although excision is primarily performed through a posterior approach [3], no specific guidelines exist for the surgical management of tailgut cysts. Because of the suspicion of malignancy, we chose a laparoscopic transabdominal approach even for a very low presacral cyst excision to preserve the integrity of the cystic wall. This study was conducted according to the CARE guidelines [4].

## 2. Case Report

A 48-year-old male with an unremarkable medical history was referred to our department with perianal and pelvic pain, and constipation. Physical examination revealed a mass in the rectum. Endoscopic evaluation revealed a normal rectal mucosa. Magnetic resonance imaging (MRI) confirmed the presence of a cystic mass below S3, with a solid component in the median presacral position posterior to the mesorectum. The mass extended longitudinally by 6 cm, imprinting the right posterior wall of the lower rectum near the levator ani muscles. A diagnosis of a tailgut cyst of 36 × 23 × 33 mm was made, including both a solid component with T2 contrast enhancement and multiple small cyst-like components with a maximum diameter of 10–12 mm (Figure 1). Transperitoneal laparoscopic resection was proposed, as it was deemed safer to preserve the integrity of the cyst wall.

### 2.1. Surgical Technique

Different approaches to rectal dissection have been described for benign and malignant disease [5,6]. In our technique, the patient was placed in the supine decubitus position. After laparoscopic entry with an open Veress-assisted technique [7], four trocars were placed. A circumferential incision of the peritoneal reflection was made posteriorly and laterally, saving the anterior plane. Rectal mobilization began from the posterior aspect, preserving the mesorectal fascia to visualize the cyst, which was isolated and excised to avoid injuries to the capsule. Peritoneal continuity was restored. Specimen extraction was performed via a suprapubic incision (Figure 2).

The procedure may be complicated by injury to the nerves, supplying the rectum and pelvic plexus. Furthermore, bleeding from the presacral vein must be avoided because it is difficult to manage. We suggest mobilizing the posterior wall as the initial step, in order to avoid injuries at pelvic splanchnic nerves, also known as *nervi erigentes,* which can help to preserve sexual function in male patients.

The estimated operative time was 120 min, estimated blood loss was insignificant, and no transfusion was performed. No postoperative complications occurred, and the patient was discharged on postoperative day two. No signs of recurrence were found at the six-month follow-up and no functional urinary or anorectal symptoms were reported. Preoperative management and postoperative follow-up at six months included MRI and endoscopy, which were negative for metastasis.

### 2.2. Pathology

Histopathological examination was performed on the excised specimen (5 × 4.3 × 2.5 cm), and macroscopically, the mass was partly multicystic and partly solid in appearance. The solid area corresponded to a neuroendocrine neoplasm with a nested trabecular pattern of growth comprising round cells with salt-and-pepper nuclear chromatin staining positive for chromogranin and synaptophysin, with a ki67 labelling index of 5–10% (compatible with a grade 2 NET). This neoplasm arose in a conglomerate tailgut cyst. The cystic spaces were lined with immature intestinal-type epithelium and urothelium (Figure 3).

## 3. Discussion

Tailgut cysts are rare presacral tumors derived from incomplete regression of the embryonic hindgut during embryogenesis [1]. They are frequently discovered incidentally in middle-aged women who are completely asymptomatic or present with vague abdominal symptoms, such as constipation, urinary disorders, or lower abdominal pain [8].

Specific signal intensity has been described for diagnosing tailgut cysts with accurate pelvic MRI [9]. Although tailgut cysts are benign congenital hamartomas [8], they may undergo malignant transformation, resulting in adenocarcinomas, sarcomas, or NETs [10]. Although common in the appendix [11], only 29 cases of NETs developing in a tailgut cyst have been reported in the literature [12,13].

Primary tumors arising in the presacral space include chordomas, myxopapillary ependymomas, paragangliomas, schwannomas, liposarcomas, chondrosarcomas, hemangiopericytomas, and rarely carcinoid tumors [14,15]. A retroperitoneal NET is usually a direct or metastatic spread from rectal carcinoid tumors or, more rarely, from primary presacral neoplasms [16].

Estimating the true incidence of malignant transformation of tailgut cysts is difficult and the incidence differs among studies, as the literature contains mostly single-case reports and small case series. In a recent retrospective analysis of all patients who underwent resection of tailgut cysts at the Mayo Clinic, malignant transformation was found in 8% of cases [4]. However, this single-center study seems to underestimate the real incidence. In a recent systematic review including 135 case reports and 9 case series, the risk of malignant transformation was 32.1% [8]. The increasing incidence of malignant cases may be justified by the high rate of resection performed for suspicious imaging findings.

Current surgical practice promotes complete surgical resection of the presacral mass with open or minimally invasive (such as laparoscopic or robotic) approaches despite the lack of specific guidelines [8]. No standard approach is recommended for tailgut presacral excision, with the approach consequently depending on the surgeon’s self-confidence in the pelvic region and cyst location. Different surgical approaches have been described [17].

Since the first laparoscopic excision of a retroperitoneal mass was performed twenty-seven years ago [18], the laparoscopic approach for the resection of tailgut cysts has been reserved for a few cases and confined to a few centers. The use of laparoscopy in this field increased in the “laparoscopic era” from 8.2% and 11.4% in Broccard and Mastoraki’s studies, respectively [4,8], to 74% in a study by Aubert et al. [19]. Usually, the anterior approach is indicated in tumors cranial to S3 [20]; however, in the largest series of retrorectal tumors, the anterior approach was not only performed for lesions above S3 but also for tumors above and below S3, and was mainly performed laparoscopically (74%) [19].

In the present study, complete laparoscopic excision with an anterior approach was performed as the treatment of choice, guaranteeing preservation of the pelvic plexus without entering the cysts and avoiding rectal injury. For upper abdominal tumors [21,22,23], minimally invasive surgery for retrorectal tumors allows safe resection of the tumor, reducing bleeding and injury [24], and improving recovery [25].

In conclusion, the present case report could help establish more well-grounded recommendations in the near future regarding the surgical management of tailgut cysts, promoting the use of a minimally invasive anterior approach. Because a tailgut cyst can conceal a malignant tumor, it is necessary to preserve its integrity. We demonstrated that in selected cases, even for very few lesions, the laparoscopic transabdominal technique may be safe and feasible.

## Figures and Tables

**Figure 1 healthcare-10-00805-f001:**
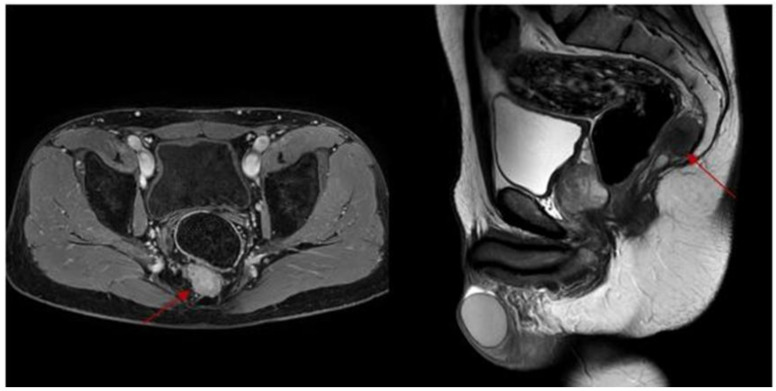
Magnetic resonance imaging shows the tailgut cyst in the presacral space below S3.

**Figure 2 healthcare-10-00805-f002:**
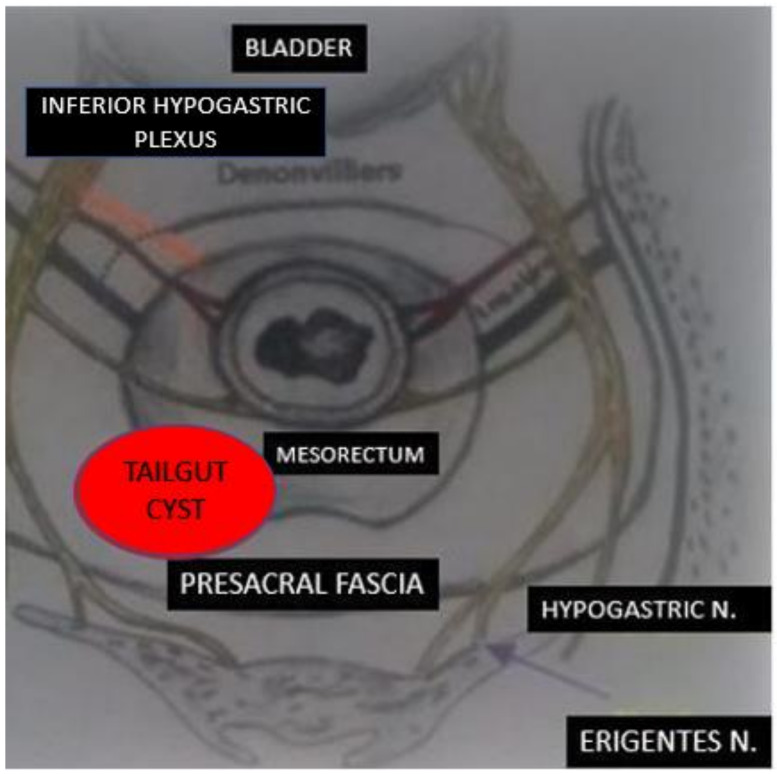
Location of this cystic mass and surrounding organs.

**Figure 3 healthcare-10-00805-f003:**
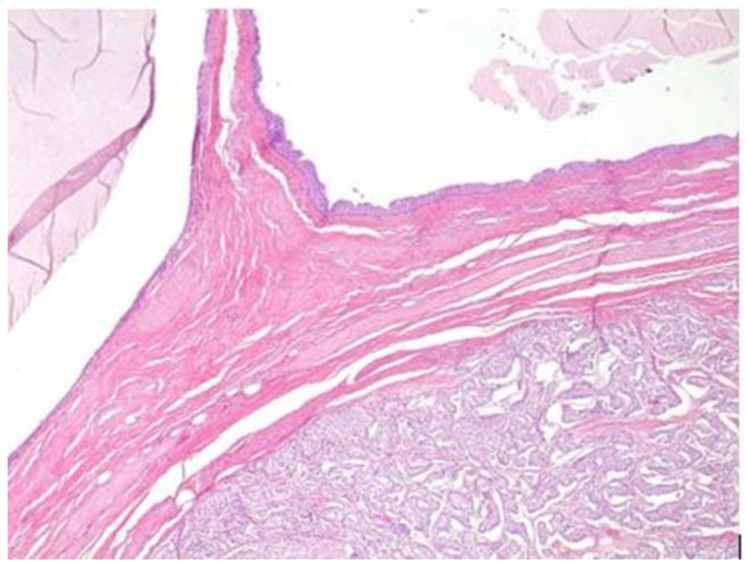
Histological examination: An area of the multiloculated cyst with an immature, mucus-secreting intestinal epithelial lining below the two lumens. The neuroendocrine neoplasm consists of nests and bundles of small monomorphic cells with a characteristic organoid growth pattern (hematoxylin and eosin 5× HPF).

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
