# Peer review of "Promoting Laparoscopic Anterior Approach for a Very Low Presacral Primary Neuroendocrine Tumor Arising in a Tailgut Cyst"

_healthcare, 2022, doi:10.3390/healthcare10050805_

Round 1

Reviewer 1 Report

The manuscript entitled “Promoting laparoscopic anterior approach for a very low presacral primary neuroendocrine tumour arising in a tailgut cyst” by Maria Michela Di Nuzzo, et al. reported that this manuscript may help to better define more solid recommendations about the surgical management of retro rectal tumors demonstrating that the laparoscopic transabdominal technique is safe and feasible even for lesions below the third sacral vertebral body. Although the study is of interest and important, there are some flaws to be revealed.

Major comments

  1. The authors should create a schema of surgical procedure in SURGICAL TECHNIQUE session to clarify the location of this cystic mass and surrounding organs.
  2. The authors should add some of laparoscopic figures to clarify the usefulness of laparoscopic transabdominal approach.
  3. The authors revealed solid recommendations about the surgical management of tailgut cysts strengthening the use of minimally invasive anterior approach. I think it would be better to add some surgical pitfalls or knacks in this procedure.
  4. The author mentioned this tumor is a "rare primary neuroendocrine tumour arising in a tailgut cyst". However, the authors need to deny metastasis using pre- or postoperative data (e.g., octreotide scans, GI tract endoscopy including small intestine, and 18F-FDG/PET-CT).

Minor comments

  1. The authors should unify “primary neuroendocrine tumour” and “malignant tumor” in Title and Background.
  2. The authors should correct “No signs” to “no signs” in Case report session of Abstract.
  3. The authors should correct “elevator ani muscles” to “levator ani muscles” in CASE REPORT session.
  4. Some grammatical errors are found in the main manuscript. It should be reviewed by a native English editor.

Author Response

file rev1 

Reviewer 2 Report

This manuscript describes a case of a tailgut tumor in a 48-year old male managed with a laparoscopy. The post-operative observation time was 6 months. The Authors conclude, the paper is another proof that laparoscopic approach is a feasible management of medium sized tailgut tumors.

Besides the manuscript language requiring much of improvement, this paper brings a little of new insight into a surgical treatment of this pathology. An extensive comparison of abdominal (laparoscopic) and posterior (open surgery) approach has recently (2021) been published on a large number of 270 cases, the paper was quoted (cit. 19) by the Authors themselves.

I do not recommend publishing this manuscript in 'Healthcare' because of the paucity of data and an insufficient language.

Author Response

file rev2

Reviewer 3 Report

The authors present a case report concerning laparoscopic resection of a very low presacral tailgut cyst with primary NET. The case is well presented and discussed. Unfortunately, the statement that the laparoscopic transabdominal technique is safe and feasible even for lesions below the third sacral vertebral body cannot be the conclusion of this report. 

The conclusion is that this technique was safe and feasible for you in this case

"This paper was written according to CARE guideline [3]." - citation is incorrect

Author Response

file rev  3

Round 2

Reviewer 1 Report

It is almost acceptable.

The authors should correct "Pelvic plexus" in Figure 2 because it is garbled.

Author Response

Dear Editor and Reviewers,

Thank you for your helpful comments and advice provided to improve the quality of our manuscript :

Promoting laparoscopic anterior approach for a very low presacral primary neuroendocrine tumour arising in a tailgut cyst.

Please find the comments and the list of changes we have made below.

Reviewer 1

1.The authors should correct "Pelvic plexus" in Figure 2 because it is garbled.

Reply 1:

Thank you for your comment. We have modified as requested.

Reviewer 2 Report

The language of the manuscript has been improved.

An additional figure was included to better describe anatomical relations of parasympathic nerves. In its new form, this paper may focus operators' attention to the presence of pelvic splanchnic nerves, also known as nervi erigentes, an important hint helping to preserve sexual function in male patients.

In they reply, the Authors underline, that their approach deviates from standard surgical practice.

Author Response

Dear Editor and Reviewers,

Thank you for your helpful comments and advice provided to improve the quality of our manuscript :

Promoting laparoscopic anterior approach for a very low presacral primary neuroendocrine tumour arising in a tailgut cyst.

Please find the comments and the list of changes we have made below.

Reviewer 2

1.The language of the manuscript has been improved.

An additional figure was included to better describe anatomical relations of parasympathic nerves. In its new form, this paper may focus operators' attention to the presence of pelvic splanchnic nerves, also known as nervi erigentes, an important hint helping to preserve sexual function in male patients.

In they reply, the Authors underline, that their approach deviates from standard surgical practice.

Reply 1: Thank you for your suggestion, we add this aspect in the text as follows:

 "We suggest mobilizing the posterior wall as the initial step , in order to avoid injuries at pelvic splanchnic nerves, also known as nervi erigentes, an important hint helping to preserve sexual function in male patients."